# Evaluation of the Leachability of Contaminations of Fly Ash and Bottom Ash from the Combustion of Solid Municipal Waste before and after Stabilization Process

**Monika Czop** [1,*] and **Beata Łaźniewska-Piekarczyk** [2]

1   Department of Technologies and Installations for Waste Management, Faculty of Energy and Environmental Engineering, The Silesian University of Technology, Konarskiego 18, 44-100 Gliwice, Poland
2   Department of Building Materials and Process Engineering, Faculty of Civil Engineering, The Silesian University of Technology, Akademicka 5, 44-100 Gliwice, Poland; beata.lazniewska@polsl.pl
*   Correspondence: monika.czop@polsl.pl; Tel.: +48-3-2237-2104

**Abstract:** The aim of this work was to check the possibility of using a concrete matrix to immobilize contaminants from ash (fly and bottom) originating from the combustion of solid municipal waste. This work presents tests of ash from a Polish incineration plant. Nowadays, the management of ash poses a big problem with respect to the high concentration of contaminants that constitutes an environmental nuisance (heavy metals, chlorides, sulfates, etc.). The excessive leaching of contaminants disqualifies ash from being deposited in landfills for hazardous wastes. Bottom ash following the combustion of solid municipal waste mainly contains calcium (23.81%), chlorine (5.44%) and heavy metal ($\Sigma$ 11.27 g/kg) compounds, while fly ash is characterized by a high content of chlorine (7.22%) and heavy metals ($\Sigma$ 7.83 g/kg). In the next stage, two concrete mixtures were designed and prepared, containing 30% of ash from combustion of solid municipal waste. The most advantageous physicomechanical properties had concrete mortars that contained 30% of bottom ash: compressive strength—29.48 MPa, bending strength—1678 kN. The performed tests showed that immobilization of dangerous compounds through the C-S-H phase of the concrete significantly decreases the migration of dangerous substance into the environment and minimizes its toxicity. Approximately 97% of the chloride and sulfate salt content was immobilized, and the heavy metal content was immobilized by the C-S-H phase to a degree of 90%. The results obtained provide the option of conveniently managing dangerous wastes with the use of a tight and durable concrete. In many cases, such technology may constitute the best and the cheapest long-term solution in the waste management economy. It may also fill a market gap in this field.

**Keywords:** municipal solid waste; hazardous waste; minimization; utilization; immobilization

## 1. Introduction

The waste management system is a multifaceted structure whose operation should ensure sustainable development in three aspects: economic, environmental and social. The Waste Framework Directive 2008/98/WE sets out a clear hierarchy of waste management: reduce, reuse, recycle, recover and dispose [1,2]. Note that this hierarchy must be applied flexibly. Two criteria should always be taken into account, i.e., economic efficiency and the impact of the adopted method on the environment throughout the full life cycle. In 2016, the total waste generated in the EU-28 by all economic activities and households amounted to 2.5 billion tons [2]. The municipal fraction accounts for 8% of the total waste stream generated annually in the EU-28. Solid municipal waste is a big problem in

many countries, because it is visible and has a complex composition, many sources of origin, and is related to the consumption patterns in a given society. 47% of all solid municipal waste generated in the EU-28 is recycled or composted [2]. However, waste management practices differ in different countries. In several EU countries, landfilling is still the basic method of municipal waste management. In countries such as Estonia, Luxembourg, France, Ireland, Slovenia, Italy, Great Britain, Lithuania and Poland, approximately 1/3 of municipal waste generated is stored, and more than 40% (except in Estonia) is recycled and subjected to thermal processing [2].

Thermal processing is one of the elements of the current municipal waste management. Waste that cannot be recycled for various reasons is used for the production of electricity and heat. Additional advantages of thermal processing of municipal waste include: reduction of volume by 90% and weight of waste by 70% and destruction of pathogens dangerous for human health [3–5]. As a result of the thermal processing of municipal waste, secondary waste (fly and bottom ash, slag, dust from the dust removal system, etc.) is generated, which requires further management due to the negative environmental impact. The chemical composition of ashes resulting from the thermal degradation of municipal waste differs significantly from the composition of ashes generated as a result of the combustion of 'clean' coal or biomass. Compounds of alkali metals and so-called acid compounds (sulfur, chlorine) present in the waste are emitted to the atmosphere in fly ash, while heavy metals, silicon or calcium deposited in bottom ash. Fly and bottom ash from thermal processing of municipal waste are often classified as hazardous waste [3–5].

There are several ways to handle this type of waste, including storage of hazardous waste in the landfill, depositing in closed excavations of potassium salt, combustion in hazardous waste incineration plants and stabilization/solidification [6–8]. Some types of secondary waste, e.g., slags, are used in construction. The development of technology aims at a wide use of mineral binders for the disposal of hazardous waste through the solidification process (immobilization). An analysis of the literature proves that the immobilization of dangerous method, in many regions should be used as the basic method of management of this type of wastes. Immobilization is recommended due to its relatively low cost, a possibility to transform many types of waste, which may be neutralized with this method (different chemical content) and the fact that solidified/stabilized waste do not pose any threat for the environment and people's health and they may be used in the industry [8]. According to the information written in the literature, the dangerous waste which most often undergoes a process of immobilization is industrial dust and sludge, gravels and ashes from the thermal process (including iron and steelworks processing, non-ferrous metal processing, municipal waste incineration plants, etc.). Solidification/stabilization technologies may be divided into 6 groups depending on the main components used and processes applied: cement-based, lime-based, thermoplastic processes, based on organic polymers, based on hermitization, and melting processes (grass). These groups can be applied in different ways and have different requirements for initial waste processing. However, all the groups aim at achieving physicochemical properties of the wastes such that the following goals of the process can be achieved: decreasing migration of the pollution into the environment, production of unified concrete matrixes that can be economically taken advantage of and facilitation of disposal, and transport of the wastes to the landfill. The literature sources mention that the stabilization of ash from municipal waste incineration plants, without preliminary processing, in concrete matrixes does not guarantee a decrease of leachability of chlorides and sulfates to the required permissible values. Furthermore, the content of chloride and sulfate salts may adversely affect the durability of concrete matrixes [8–10]. The immobilization (solidification) process makes it possible to change the physical and chemical properties of the waste, as well as to reduce the solubility and leachability of substances harmful to the natural environment. To permanently immobilize waste in concrete, cement is commonly used, in accordance with standard requirements [9,10]. Solidification matrixes for hazardous waste should have good physio-mechanical properties, which consequently affects the durability of composites over a long period of time. The durability of composites solidifying waste is

important from an environmental point of view because it affects the leaching of contaminants into the aquatic or soil environment with which these matrices are in direct contact [9,10].

Concrete solidifying of hazardous waste stored in the environment may be exposed to various adverse environmental conditions (temperatures, precipitation, chemically aggressive groundwater and other aggressive liquids). The literature [11–18] contains little information on the impact of ashes from the municipal waste incineration plant on the properties of cement mortars as this is a novelty in the discussed problem. This article focuses on assessing the effectiveness of the process of immobilizing hazardous waste (ashes) by analyzing the leaching of contaminants (aggressive ions and heavy metals) from monolithic and crushed concretes. The purpose of the research performed was to design an environmentally friendly concrete mix and to use ash as an alternative aggregate in accordance with the idea of the Circular Economy. The performed tests demonstrate that the high resistance of the concrete makes it possible to use this type of waste for example for protecting waste landfill, where it may form a layer that separates the waste from the environment. Furthermore, the suggested solution will contribute to the decrease of greenhouse gases, $CO_2$ in particular, as the cement is replaced with waste. Moreover, the generated waste will be used in their place of origin which will eliminate the economic and environmental costs of international transport.

## 2. The Analyzed Installation of Thermal Processing of Municipal Waste

In the analyzed installation of thermal processing of municipal waste, in order to increase reliability, there are 2 combustion lines that process 210 Mg of waste per year (Figure 1). The prevailing temperature in the boiler is 1000 °C, and the cleaned gases are discharged by 50 m high chimneys. The effect of the process is electricity in the amount of 128,000 MWh/year and heat energy in the amount of 300,000 GJ/year. All energy produced is used for the needs of the city [19].

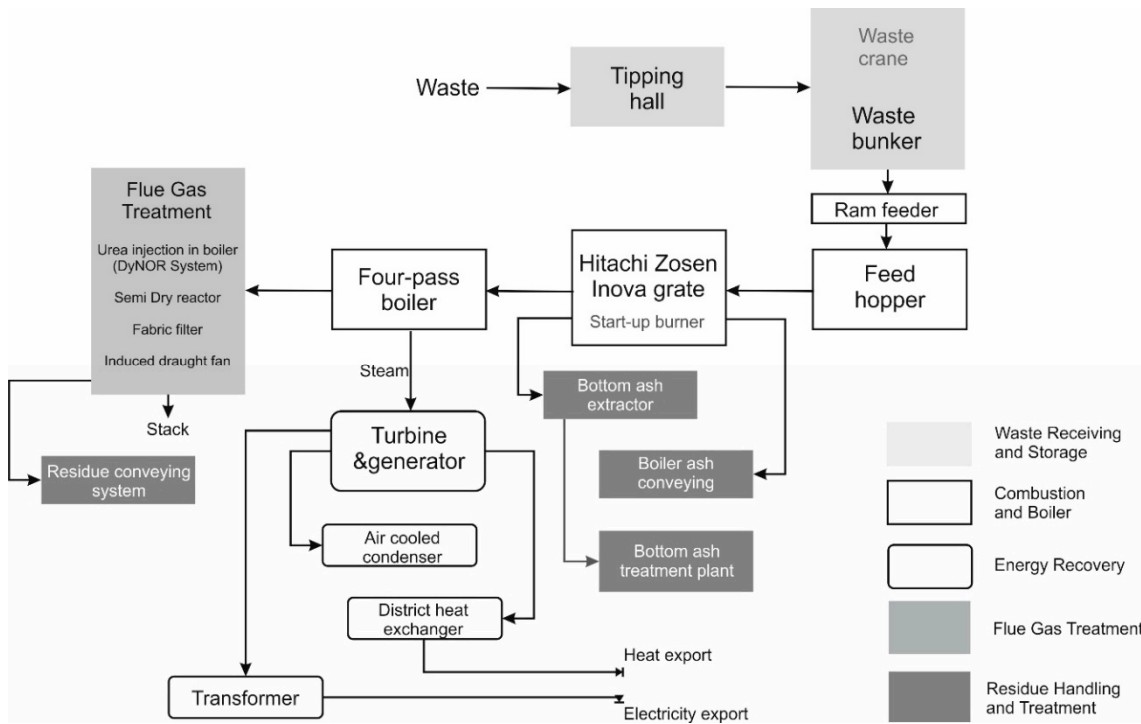

**Figure 1.** Simplified scheme of an analyzed MSW incinerator.

The installation receives waste that is not suitable for recycling or for other forms of management, including residual waste, as its average calorific value is 7.5 GJ/Mg. The waste constitutes, among other things: mixed municipal waste (20 03 01); bulky waste (20 03 07), mainly including used or damaged furniture; waste which, after passing through the sorting plant, has not been classified as

secondary raw material; and so-called other wastes (including mixed substances and objects) from mechanical processing of wastes other than those mentioned in 19 12 11 (19 12 12) and, additionally, wastes constituting combustible waste (refuse-derived fuel) (19 12 10) [20].

Municipal waste collected in the city is transported by specialized cars for installation. At the site, the waste is collected and stored. The car enters the scale, where the mass of the collected waste is measured, and passes through the radiosensitive gate. Then the car is transported to the delivery hall, where there is a negative pressure in order not to spread unpleasant odors. The waste is placed in a bunker, where the gripper on the crane mixes the waste to avoid self-ignition and uniformity of mass in case the plant stops, which happens at least once a year. A separate independent deodorizing system is installed in the bunker. Waste prepared in this way is ready for thermal processing and energy recovery. Using a gripper, the waste is placed in the hopper; then, the feeder is directed to the grate, where, in cooperation with the auxiliary burners, they are thermally processed. The exhaust gases generated in the process are cleaned [19]. The exhaust gas cleaning system in the installation consists of the following elements: bag filters, NOx reduction system, semi-dry reactor, draft fan as well as chimney and emission measurement station. A turbine-generator, air-cooled condenser and heat exchanger are installed for energy recovery. The post-processed waste is: slag (19 01 11), and bottom ash and fly ash from the exhaust gas process (19 01 07 *). 3472 Mg of ferrous metals and 624 Mg of non-ferrous metals are recovered from wastes subjected to thermal processes [19].

## 3. Materials and Methods

### 3.1. Materials

For this research, fly ash (Figure 2a) and bottom ash (Figure 2b) were used. The waste is generated in the process of thermal degradation of municipal wastes. In accordance with the binding Waste Catalogue [11], the tested ashes (fly and bottom) are classified under code 19 01 07 *—solid wastes from gas treatment. The tested ashes, due to the high content of substances harmful to the environment, are classified as hazardous waste. The ashes were light grey, had a heterogeneous grain size. It should be added that the fly ash was characterized by an irritating, pungent odor. Currently, the tested ashes are neutralized by storage in inactive salt, potassium and magnesium mines.

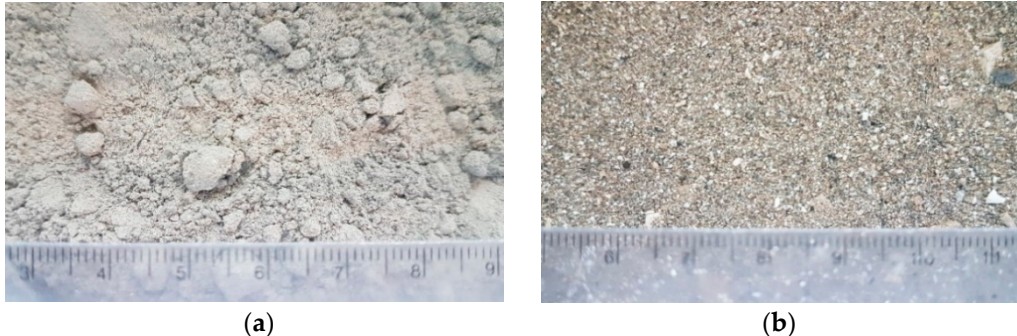

|        (a)        |        (b)        |

**Figure 2.** Analyzed ashes: (**a**) fly ash (FA), (**b**) bottom ash (BA) (author: Monika Czop).

The second material was concrete beams (Figure 3) made of CEM Portland cement with 30% ash addition. The prepared concrete beams had the following dimensions: $40 \times 40 \times 160$ mm. The composition of the concrete mixture was as follows: CEM I Portland cement/binder (fly or bottom ash), standard sand, and water. The water/cement ratio (w/c) in all concrete mixtures was 0.5. The concrete beams with the addition of the tested ashes were matured in the laboratory for 28 days.

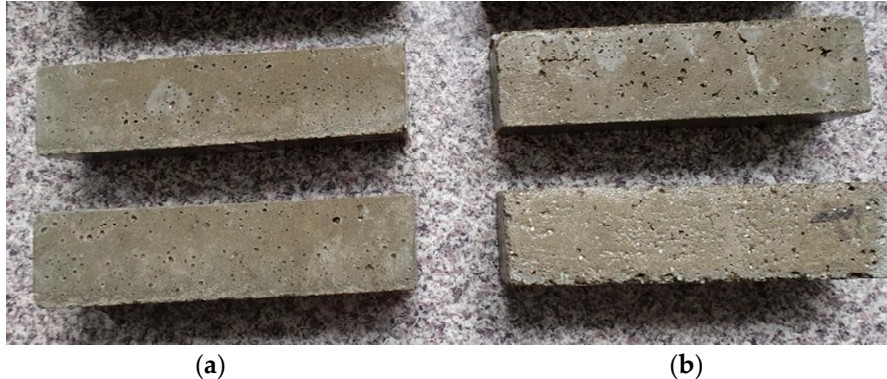

(**a**)          (**b**)

**Figure 3.** Tested concrete sample (**a**) with 30% admixture of fly ash, and (**b**) with 30% admixture of bottom ash (author: Beata Łaźniewska-Piekarczyk).

### 3.2. Methods

The research procedure consisted of five stages:

- Testing physicochemical properties of ash (fly and bottom) that is hazardous waste,
- Preparation of water extracts from ashes and assessing their degree of environmental nuisance,
- Preparation of water extracts from monolithic concretes after 28 days of maturation, performing chemical tests and assessing the degree of immobilization of contaminants,
- Preparation of water extracts from crushed concretes after 28 days of maturation, performing chemical tests and assessing the degree of immobilization of contaminants,
- Tests of compressive strength and bending strength of the designed concrete mortar with an admixture of fly ash and bottom ash as compared with the reference tests.

In the ash samples (fly and bottom) submitted for testing, the water content was determined based on the PN-EN 15934: 2013 [21] standard, and bulk density was determined in accordance with the PN-EN 1097-3:2000 standard.

To obtain research material with an appropriate degree of homogenization, the tested ash samples were subjected to preliminary grinding in a ball mill. The ash samples prepared in this way were subjected to selected physicochemical analyses. The specific surface area was determined according to the PN-EN 196-6:2019-01 standard, and loss on ignition of dry matter in accordance with the PN-EN 15935:2013-02 standard and the PN-EN 196-2:2013-11 standard.

The samples were also tested for the following elements: carbon (C) PN-EN 15407:2011 [22], organic carbon (TOC) PN-Z-15011-3:2001, nitrogen (N) PN-G-04523:1992, sulfur (S) PN-ISO 334:1997 [23] and chlorine (Cl) PN-ISO 587:2000 [24]. Determination of sodium, calcium, potassium, lithium and barium content in ashes was determined by flame emission spectrometry—in accordance with PN-ISO 9964-3:1994 [25]. Inductively coupled plasma mass spectrometer (ICP MS) from Perkin Elmer was used to assess heavy metal content in the dry matter of samples, which allows determination of elements coupled in an argon plasma.

Execution of water extracts from ashes was carried out according to PN-EN 12457-2:2006 standard [26]. A representative laboratory sample, which weighed 2 kg, was prepared from the collected waste. For analysis, the tested waste was sieved through 2 mm sieves, and from the sample prepared in this way, the water extract was prepared at a ratio of liquid to solid phase L/S = 10 dm$^3$/kg (basic test). The elution liquid was distilled water with pH 7.1 [27] and an electric conductivity of 61.18 μS/cm [28]. The extract was then shaken in a laboratory shaker for 24 h before the suspension was filtered.

In the same way, water extracts were prepared for concrete samples (after 28 days of maturation), both monolithic and ground to grain sizes <10 mm [20,29].

The analysis of the water extracts from the ashes and concretes included several determinations. Organic carbon content was determined using Elementar's Vario TOC Cube analyzer [30]. Solution pH and conductivity were determined using the Elmetron CPC-501 apparatus. Chloride content was determined using the Mohr method, with silver nitrate (V) as a titration reagent and potassium chromate (VI) as an indicator (PN-ISO 9297:1994 [31]). Determination of sulfates (VI) ($SO_4^{2-}$) was carried out by the gravimetric method with barium chloride according to the PN-ISO 9280:2002 standard [32]. The determination of of sodium, calcium, potassium, lithium and barium contents in the water extracts from the ashes and concretes was performed by flame emission spectrometry, in accordance with PN-ISO 9964-3:1994 standard [25,33]. An inductively coupled plasma mass spectrometer (ICP MS) from Perkin Elmer was used to assess the heavy metal content of the water extract, thus allowing the determination of elements coupled in the argon plasma [34].

The mortars (Table 1) were prepared in accordance with the procedure described in standard PN-EN 480-1 [35,36]. The air void content in the mortar was determined as well as, for selected mortars, consistency, bending and compressive strength. Bending strength and compressive strength tests were repeated after 28 days according to standard PN-EN 196-1:2006—Concrete Test Methods. Part 1: Determination of the strength [37]. The samples were removed from the forms after 48 h and kept in water until testing. The tests were performed for maturing mortars at a temperature of 20 ± 2 °C.

**Table 1.** Composition mix of concrete mortars, expressed in grams.

| Type of Waste | Symbol of Mortar | CEM I | Water | Sand Acc. EN 196-1 |
|---|---|---|---|---|
| Reference sample from Portland cement 42.5 R | CEM I | 450 | 225 | 1350 |
| Cement + 30% bottom ash | CEM I + 30% BA | 315 | 135 | 1350 |
| Cement + 30% fly ash | CEM I + 30% FA | 315 | 135 | 1350 |
| Cement + 30% fly ash from power plants | CEM I + 30% FA_PP | 315 | 135 | 1350 |

## 4. Results and Discussion

Table 2 presents the basic technical properties of the ashes tested. The total moisture of the tested ashes was low—below 1.5%. It should be added that the fly ash tested showed hygroscopic properties. In laboratory conditions, a water content increase of approx. 3% was observed within 7 days. The bulk density of the examined ashes was in the range (478–540) kg/m$^3$, which is lower than the density of fly ash from coal power plants (ok. 799 kg/m$^3$).

**Table 2.** Basic technical properties of tested ashes.

| Properties | Symbol | Unit | Fly Ash | Bottom Ash |
|---|---|---|---|---|
| Moisture | M | % | 1.26 | 0.70 |
| Bulk density | $\rho_b$ | kg/m$^3$ | 478.00 | 540.40 |
| Specific surface area | S | cm$^2$/g | 7454.77 | 1509.00 |
| | | m$^2$/g | 0.74 | 0.15 |

The specific surface area for the fly ash was 7454.77 cm$^2$/g, and 150.9.0 cm$^2$/g for the bottom ash, which classifies the tested ashes into a group of ashes with a developed specific surface area. This is due to their fine grain size and the occurrence of many porous, spherical grains with an extensive structure.

Figure 4 shows the results of loss on ignition (LOI) for the ashes tested. Ignition losses were determined by heating the tested ashes to constant mass in a muffle furnace at three temperatures: 600 °C, 815 °C and 950 °C in an oxidizing atmosphere.

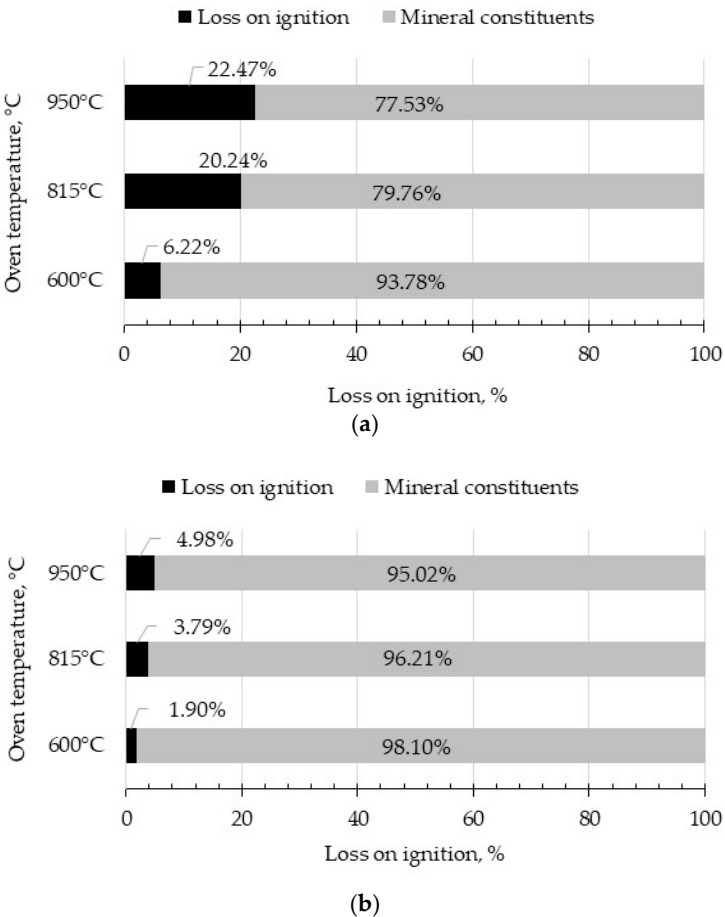

**Figure 4.** Loss on ignition at various temperatures for ashes from thermal degradation of municipal waste: (**a**) fly ash (FA); (**b**) bottom ash (BA).

In both cases, it was found that loss on ignition (LOI in 600 °C) met the criteria for admission of non-hazardous (LOI ≤ 8%) and inert waste (LOI ≤ 10%) at the landfill. In addition, ash loss on ignition was determined by ignited samples at 950 °C for an extended time of up to 1 h. This parameter is important because of the production of concrete blocks for the immobilization of contaminants. Fly ash due to the loss on ignition is divided into three categories: category A (LOI ≤ 5%), category B (LOI ≤ 7%), and category C (LOI ≤ 9%). Based on literature data, it is recommended to use category A ash for concrete, as high losses on ignition in ashes may result in deterioration of workability of the concrete mixture. Fly ash may be problematic because of its workability as its losses on ignition were above 20%.

Table 3 presents the chemical composition of the ashes tested. Total carbon in the tested ashes was below 7%. Particular attention should be paid to the content of organic carbon, as excessive amounts cause a decrease in the effectiveness of chemical admixtures, especially aeration agents, plasticizers and superplasticizers. In addition, the pozzolanic activity also decreases, affecting the unsightly appearance of the concrete surface and hindering the process of surface hardening of the concrete. The organic carbon content is a maximum of 3% of the ash mass. In the case of bottom ash, the total sulfur constitutes 1.41% of the waste mass, and for the fly ash, this value is 0.66%. The tested ashes had a high Ca content; in fly ash, this was at the level of 15.7%, while in the bottom ash, the value was 23.81%. Calcium content is associated with the waste gas processing system. Particular attention should be paid to the chlorine content in the ashes tested. The fly ash had a chlorine content of 7.22%, while the chlorine content in the bottom ash was 5.44%. Chlorine is an undesirable and environmentally troublesome element.

**Table 3.** Chemical composition of tested ashes from combustion of the solid municipal waste.

| Properties | Symbol | Unit | Fly Ash | Bottom Ash |
|---|---|---|---|---|
| Total carbon | C | % | 6.42 | 1.86 |
| Total organic carbon | TOC | % | 2.52 | 0.25 |
| Nitrogen | N | % | 0.35 | 0.22 |
| Sulfur | S | % | 0.66 | 1.41 |
| Chlorine | Cl | % | 7.22 | 5.44 |
| Orthophosphate (V) | $P_2O_5$ | % | 1.32 | 0.96 |
| Phosphorus | P | % | 0.57 | 0.42 |
| Potassium | K | % | 1.69 | 12.72 |
| Calcium | Ca | % | 15.70 | 23.81 |
| Bar | Ba | % | 0.48 | 4.01 |
| Lithium | Li | % | 0.02 | 0.07 |
| Sodium | Na | % | 1.41 | 10.28 |

The heavy metal content in the fly and bottom ash formed during the process of thermal degradation of municipal waste had the following sequence: Zn > Pb > Cu > Cd. The tested ashes had a heavy metal content in the range presented in Figure 5. Among the analyzed metals, the highest content was found for zinc, at 944.10 mg/kg in the case of bottom ash, and the lowest was found for cadmium, at 56.88 mg/kg in the case of fly ash.

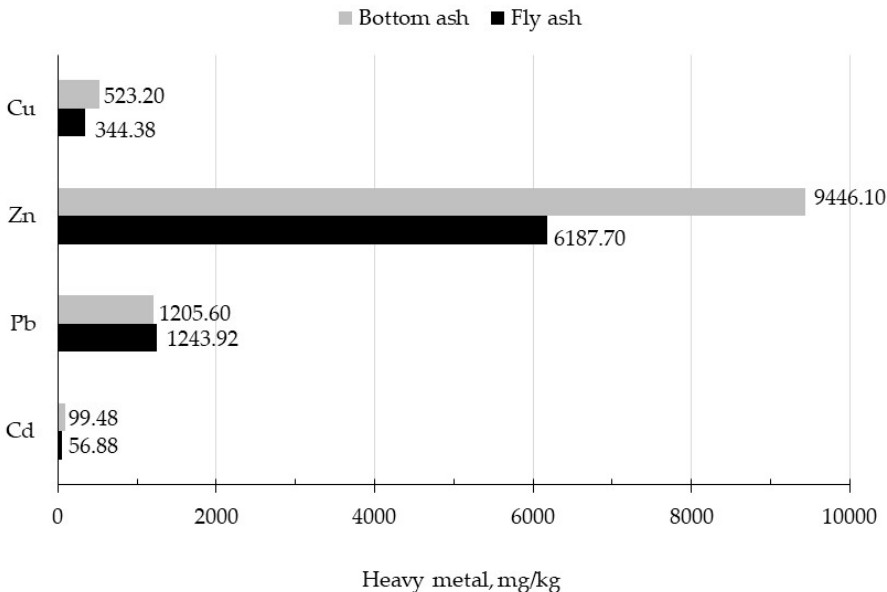

**Figure 5.** Heavy metal content in the tested ashes from the combustion of solid municipal waste.

Table 4 presents the leachability ranges of selected contaminants that may constitute an environmental problem.

**Table 4.** Leachability of selected contaminants of tested ashes, expressed in mg/kg (with pH exception).

| Properties | Symbol | Fly Ash | Bottom Ash | Criteria for Landfills [29] | |
|---|---|---|---|---|---|
| | | | | **Non-Hazardous Waste** | **Hazardous Waste** |
| pH | pH | 12.0 | 9.8 | min. 6 | - |
| Total Carbon | TC | 1578.5 | 310.9 | - | - |
| Total Organic Carbon | TOC | 1578.5 | 252.4 | - | - |
| Total Inorganic Carbon | TIC | blq * | 58.5 | - | - |
| Chloride | $Cl^-$ | 91,584.0 | 73,728.0 | 15,000 | 25,000 |
| Sulfate | $SO_4^-$ | 8200.0 | 6552.2 | 20,000 | 50,000 |
| Phosphate trianion | $PO_4^-$ | 65.7 | 52.8 | - | - |
| Phosphorus | P | 21.4 | 17.2 | - | - |
| Potassium | K | 16,898.3 | 23,310.0 | - | - |
| Calcium | Ca | 24,995.0 | 8940.0 | - | - |
| Lithium | Li | 110.0 | 40.0 | - | - |
| Sodium | Na | 13,383.3 | 17,970.0 | - | - |
| The sum of chloride and sulfate | TDS | 99,784.0 | 80,280.2 | 60,000 | 100,000 |

* blq—Values below the limit of quantification.

High pH values for the tested ashes (pH > 9) may indicate high immobilization of heavy metals in the material. This is confirmed by the low concentrations of the studied metals in water extracts (Table 5). The analyzed ashes can be an environmental problem due to their high salt content, mainly in the case of chloride and sulfate salts. The chloride leaching from both tested ashes exceeded the permissible levels for storage at hazardous waste landfills. The level of sulfate leaching was high but did not exceed the permissible standards.

**Table 5.** Content of heavy metals in water extracts from the ashes tested, expressed in mg/kg.

| Properties | Symbol | Fly Ash | Bottom Ash | Criteria for Landfills [29] | |
|---|---|---|---|---|---|
| | | | | **Non-Hazardous Waste** | **Hazardous Waste** |
| Bar | Ba | 5461.7 | 8750.0 | 100 | 300 |
| Zinc | Zn | 6.72 | blq * | 50 | 200 |
| Copper | Cu | 0.04 | blq * | 50 | 100 |
| Lead | Pb | 0.65 | 0.63 | 10 | 50 |
| Cadmium | Cd | blq * | 0.01 | 1 | 5 |
| Chrome | Cr | blq * | 0.70 | 10 | 70 |
| Cobalt | Co | blq * | blq * | - | - |
| Iron | Fe | blq * | blq * | - | - |
| Manganese | Mn | blq * | blq * | - | - |
| Nickel | Ni | 0.16 | 1.51 | 10 | 40 |

* blq—Values below the limit of quantification.

The heavy metal content, i.e., barium, zinc, lead, copper, cadmium, chromium, cobalt, iron, manganese and nickel, was determined in the water extracts obtained from the ashes tested (Table 5). From the fly ash, Ba was leached to the greatest extent, followed by Zn > Pb > Ni > Cu; the content of other metals (Cd, Cr, Co, Fe, Mn) was below the limit of quantification. Meanwhile, in the case of the bottom ash, Ba was leached the most, and then Ni > Cr > Pb > Cd, and the content of metals such as Zn, Cu, Co, Fe, Mn was below the limit of quantification. Due to the high content of barium, the ashes tested did not meet the criterion for being stored in landfills for non-hazardous and inert waste, or even for hazardous waste. The barium content exceedances were very high.

In parallel with the physicochemical research on ashes, concrete blocks with 30% waste added were designed and manufactured. The manufactured concrete blocks were matured for 28 days in laboratory conditions, after which water extracts were made of them. The leachability of contaminants from concrete with the addition of hazardous waste ashes may be related to the form in which this concrete occurs (monolithic or comminuted form), and this may translate into an environmental

nuisance. In the case of monolithic forms, the level of leaching may mainly be determined by the process of release from the surface and diffusion.

In the case of crushed concrete, the leaching of contaminants determines the percolation process. This article presents the leachability of contaminants harmful to the environment for both forms of concrete. Both cases are analyzed, and the results obtained are presented later in the article.

Table 6 presents the leachability of selected contaminants from monolithic concrete that can potentially be a nuisance to the natural environment. High pH values (pH = 11) may indicate high immobilization of heavy metals and chloride and sulfate salts. The leachability of chloride and sulfate salts does not exceed the permissible levels for the storage of waste in landfills other than those for hazardous and inert waste. Based on this research and the results obtained, it can be determined that the ashes subjected to the immobilization process in the concrete matrix were not harmful to the environment. This is also confirmed by the low concentrations of tested metals in the water extracts (Table 7).

**Table 6.** Leachability of selected contaminants from monolithic concrete, expressed in mg/kg (with pH exception).

| Properties | Symbol | Fly Ash | Bottom Ash | Criteria for Landfills [29] | |
| | | | | Non-Hazardous Waste | Hazardous Waste |
|---|---|---|---|---|---|
| pH | pH | 11.0 | 11.0 | min. 6 | - |
| Total Carbon | TC | 42.2 | 134.6 | - | - |
| Total Organic Carbon | TOC | 22.7 | 37.2 | - | - |
| Total Inorganic Carbon | TIC | 19.5 | 97.4 | - | - |
| Chloride | $Cl^-$ | 1267.2 | 138.24 | 15,000 | 25,000 |
| Sulfate | $SO_4^-$ | 106.55 | 353.80 | 20,000 | 50,000 |
| Phosphate trianion | $PO_4^-$ | 34.67 | 22.33 | - | - |
| Potassium | K | 3.27 | 5.43 | - | - |
| Calcium | Ca | 21.88 | 2.12 | - | - |
| Lithium | Li | 0.10 | blq * | - | - |
| Sodium | Na | 3.11 | 3.58 | - | - |
| The sum of chlorides and sulfates | TDS | 1373.75 | 492.04 | 60,000 | 100,000 |

* blq—Values below the limit of quantification.

**Table 7.** Leachability of heavy metals from monolithic concrete (integral), expressed in mg/kg.

| Properties | Symbol | Fly Ash | Bottom Ash | Criteria for Landfills [29] | |
| | | | | Non-Hazardous Waste | Hazardous Waste |
|---|---|---|---|---|---|
| Bar | Ba | 2.44 | blq * | 100 | 300 |
| Zinc | Zn | blq * | blq * | 50 | 200 |
| Copper | Cu | 0.002 | blq * | 50 | 100 |
| Lead | Pb | 0.06 | 0.03 | 10 | 50 |
| Cadmium | Cd | blq * | blq * | 1 | 5 |
| Chrome | Cr | blq * | blq * | 10 | 70 |
| Cobalt | Co | blq * | blq * | - | - |
| Iron | Fe | blq * | blq * | - | - |
| Manganese | Mn | blq * | blq * | - | - |
| Nickel | Ni | blq * | 0.12 | 10 | 40 |

* blq—Values below the limit of quantification.

The leachability of heavy metals from monolithic concrete forms after 28 days of maturation is presented in Table 7.

The obtained concentrations of heavy metals in water extracts were compared with the permissible content for waste intended for storage in non-hazardous and inert landfills and hazardous waste. It is worth emphasizing that no content of heavy metals leached from monolithic concrete with 30% ash addition exceeded the permissible storage values for either considered case. The content of heavy

metals in water extract from monolithic concretes was arranged in the following sequence: with the addition of fly ash, Ba > Pb > Cu; and with the addition of bottom ash, Ni > Pb.

In the course of the tests carried out, the crushed concrete was also leached. Table 8 shows the content of selected contaminants that can be harmful to the environment in the water extracts from crushed concrete after 28 days of maturation.

**Table 8.** Leachability of selected contaminants from crushed concrete, expressed in mg/kg (with pH exception).

| Properties | Symbol | Fly Ash | Bottom Ash | Criteria for Landfills [29] | |
|---|---|---|---|---|---|
| | | | | Non-Hazardous Waste | Hazardous Waste |
| pH | pH | 12.7 | 12.5 | min. 6 | - |
| Total Carbon | TC | 42.40 | 404.60 | - | - |
| Total Organic Carbon | TOC | 26.80 | 24.20 | - | - |
| Total Inorganic Carbon | TIC | 15.70 | 380.40 | - | - |
| Chlorides | $Cl^-$ | 2027.50 | 1244.20 | 15,000 | 25,000 |
| Sulfate | $SO_4^-$ | 259.18 | 193.36 | 20,000 | 50,000 |
| Phosphate trianion | $PO_4^-$ | 59.00 | 23.33 | - | - |
| Potassium | K | 299.13 | 130.20 | - | - |
| Calcium | Ca | 590.43 | 1054.67 | - | - |
| Lithium | Li | 2.77 | 3.70 | - | - |
| Sodium | Na | 120.03 | 556.20 | - | - |
| The sum of chlorides and sulfates | TDS | 2286.70 | 1437.52 | 60,000 | 100,000 |

High pH values (pH > 12) may indicate high immobilization of heavy metals, as well as chloride and sulfate salts. The leachability of chloride and sulfate salts in crushed concretes more than doubled in relation to monolithic concrete; this did not exceed the permissible levels of waste storage. This is also confirmed by the low concentrations of tested metals in water extracts (Table 8).

Table 9 shows the content of heavy metals in water extracts of crushed concrete after 28 days of maturation. The obtained contents of heavy metals in water extracts were compared with the permissible contents for waste intended for storage in non-hazardous and inert landfills and hazardous waste [12]. It is worth emphasizing that none of the contents of heavy metals have leached from crushed concrete with the addition of ashes exceed the permissible values for storage at a hazardous waste landfill [12].

**Table 9.** Leachability of heavy metals from crushed concrete, expressed in mg/kg.

| Properties | Symbol | Fly Ash | Bottom Ash | Criteria for Landfills [29] | |
|---|---|---|---|---|---|
| | | | | Non-Hazardous Waste | Hazardous Waste |
| Bar | Ba | 72.83 | 241.93 | 100 | 300 |
| Zinc | Zn | blq * | blq * | 50 | 200 |
| Copper | Cu | blq * | blq * | 50 | 100 |
| Lead | Pb | 0.050 | 0.001 | 10 | 50 |
| Cadmium | Cd | 0.001 | blq * | 1 | 5 |
| Chrome | Cr | blq * | blq * | 10 | 70 |
| Cobalt | Co | blq * | blq * | - | - |
| Iron | Fe | blq * | 0.004 | - | - |
| Manganese | Mn | blq * | blq * | - | - |
| Nickel | Ni | 0.065 | 0.039 | 10 | 40 |

* blq—Values below the limit of quantification.

It should be noted that the crushed concrete with the addition of fly ash does not exceed the permissible values for storage at the landfill for non-hazardous and inert waste. Ba was marked by a high level of leaching, while in the case of crushed concrete with the addition of fly ash, it was 72.83 mg/kg; however, this content did not exceed the permissible values for storage in landfills for

either considered case. Leachability more than three times higher was recorded for crushed concrete with the addition of bottom ash, the content of which was 241.93 mg/kg. Such a situation prevents the storage of this type of concrete in landfills other than those for hazardous and inert waste. The content of heavy metals in water extracts from crushed concrete was arranged in the following sequence: with the addition of fly ash, Ba > Ni > Pb > Cd; and with the addition of bottom ash, Ba > Ni > Fe > Pb.

The concentration of heavy metals in the water extracts obtained from the monolithic concrete forms after 28 days of maturation was lower than the concentration of heavy metals in the water extracts obtained from the crushed concrete. This may be due to the fact that subsequent crushing of concrete reveals subsequent surfaces from which heavy metals can be leached.

Figures 6 and 7 present the test results for the bending and compressive strength of concrete mortars with the admixture of fly ash and bottom ash. For comparison of the obtained test results, the same trial was performed for mortar with fly ash from the power plant. The reference concrete mortar was Portland cement CEM I 42.5 R.

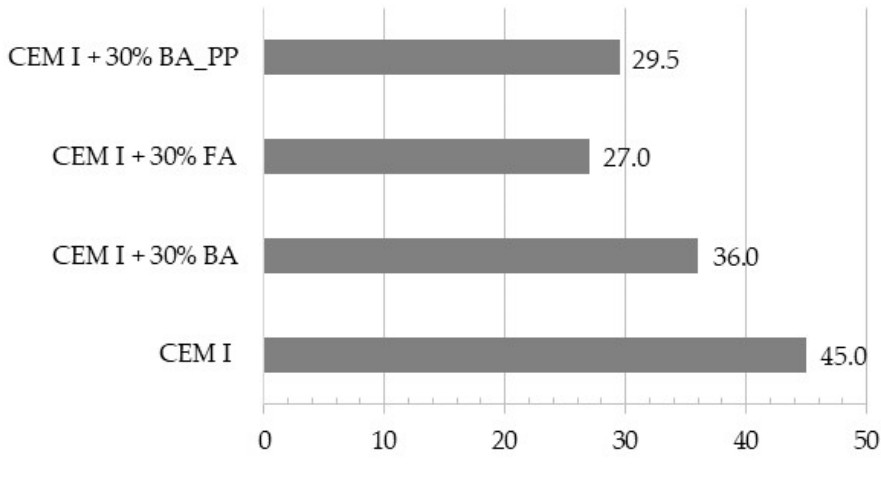

**Figure 6.** Results of compressive strength of concrete mortars after 28 days of maturation.

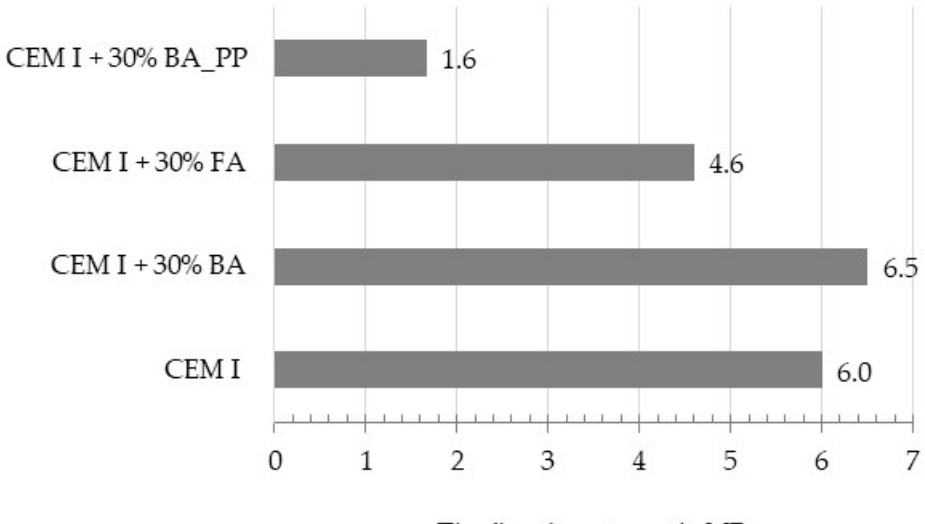

**Figure 7.** Results of bending tensile strength tests of concrete mortars after 28 days of maturation.

Analysis of the results of the performed tests showed that the tested ashes form a component that fills the microstructure of the mortars without the properties of puzzolana, similar to powder, commonly used for mortars (Figure 8).

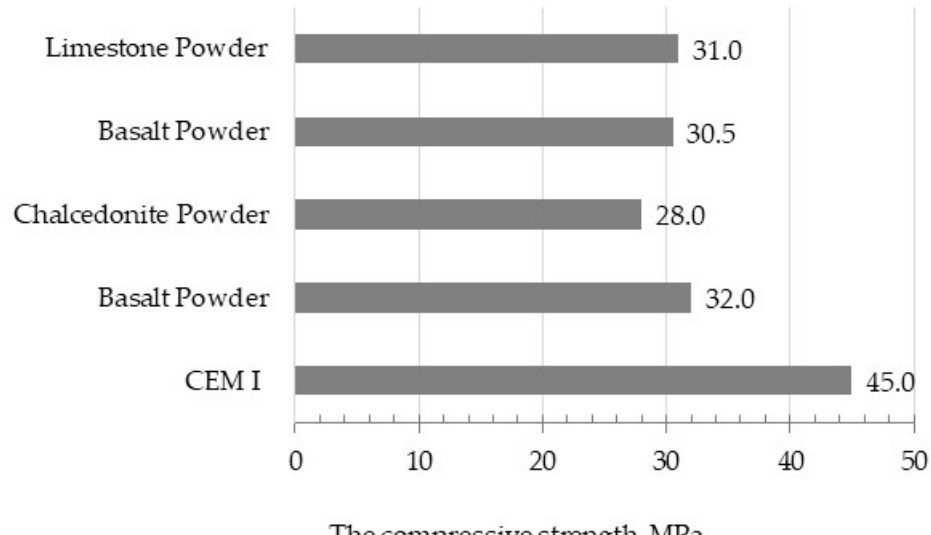

**Figure 8.** Results of compression strength tests with various types of powders after 28 days of maturation (our tests, unpublished).

The results of the presented tests prove that both fly ash and bottom can potentially be applied. Furthermore, hazardous waste that is a significant nuisance for the environment is being managed. It also has to be noted that the tested mortar did not undergo volume change and maintained its original dimensions. It can be concluded that the tested ashes, when used as a replacement for the cement, do not cause volume change of the cement binder, which is important from the point of view of the possibility of application.

The test results show unequivocally that concrete mortars with the admixture of ash (fly ash and bottom ash) are characterized by the same resistance as concrete mortars with the admixture of limestone powder, which are widely used in the construction industry. On the basis of the obtained test results, it can be concluded that the design concrete matrixes can be applied in similar technological circumstances. A search for a concrete matrix that is environmentally friendly and has good mechanical properties with the addition of ash from the thermal processing of municipal waste is still ongoing.

## 5. Conclusions

The obtained results confirm that the tested ashes, which come from the thermal degradation of municipal wastes, may be successfully neutralized in the concrete matrix. The test results show that the designed concrete matrix makes it possible to decrease the leaching of chloride and sulfate salts to the permissible values required by law. The obtained results give a basis for the use of the designed concrete matrix with the admixture of ashes for the construction of roads and other construction work in landfills. Particular attention should be given to the need for individual preparation of the concrete mix composition in accordance with the characteristics of the waste in order to achieve the most environmentally friendly matrix.

The research makes it possible to formulate the following conclusions. Solid ash does not meet the current legal requirements for storage at hazardous waste landfills. Storage in closed excavations of potassium salt mines is not possible in every European country. The tested ashes from the combustion of solid municipal waste contained a large amount of chlorine, which may affect the durability of the concrete. The tested ashes were characterized by high (0.1–10 g/kg) contents of zinc, lead and copper, and an average content (1–100 mg/kg) of cadmium in dry matter. Studies have shown that in cases of

the improper preservation and/or processing of ash, there is a high risk of migration of aggressive ions such as chlorides, as well as sulfates and heavy metals, into the environment. The content of chloride ions in the ashes exceeded 3 times the permissible value of leaching of this parameter from waste intended for storage in hazardous landfills. The obtained results confirm that immobilization is an effective process for reducing the content of contaminants in the water extract. The content of chloride in monolithic and crushed concretes has been almost completely immobilized by the C-S-H phase. The degree of immobilization exceeds 98%. Also, the content of sulfate ions is immobilized at a similar level of 96% for both forms of the concrete matrix under consideration. It is possible to permanently immobilize heavy metals (Zn, Cu, Pb, Cd, Cr, Co, Fe, Mn, Ni) in a concrete matrix. The amount of metal ions that can be leached from concrete, both monolithic and crushed, is negligible during use.

The presented results are preliminary tests in a program aimed at limiting the leaching of contaminants from secondary waste generated in the process of thermal processing of the municipal fraction. In the next research steps, the designed concrete matrix with the addition of ashes should be tested in various environmental exposure classes according to PN EN 206, also to determine whether leachability parameters change after the concrete has remained in the given exposure classes.

**Author Contributions:** Conceptualization, M.Z. and B.Ł.-P.; methodology, M.Z. and B.Ł.-P.; validation, M.Z. and B.Ł.-P.; formal analysis, M.Z. and B.Ł.-P.; writing—original draft preparation, M.Z. and B.Ł.-P.; writing—review and editing, M.Z. and B.Ł.-P.

**Funding:** This paper is a part of pedagogical Evolution program carried on at the Silesian University of Technology in EIT labelled program Innoenergy. This paper was supported by InnoEnergy and Silesian University of Technology, Gliwice, Poland.

**Conflicts of Interest:** The authors declare no conflict of interest.

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
