# Peer review of "Evaluation of the Leachability of Contaminations of Fly Ash and Bottom Ash from the Combustion of Solid Municipal Waste before and after Stabilization Process"

_sustainability, doi:10.3390/su11195384_

Round 1

Reviewer 1 Report

Reviewer's comments:

Review report for International Journal of Sustainability (sustainability-597529): Evaluation of the leachability of contaminations of fly ash and bottom ash from the combustion of solid municipal waste before and after stabilisation process

This paper presents the possibility of using a concrete matrix to immobilize contaminants from ash (fly and bottom) originating from the combustion of solid municipal wastes. The subject of the paper is interesting and relevant to the audience of the journal. The project program is well designed and executed. The authors have obtained good results, which are analyzed and in agreement with the general literature. I believe that the presented data may be interesting and useful to other researchers. The reviewer has minor corrections that authors need to address for improving the paper’s overall quality. These are given below for due attention.

Specific Comments:

P1, L25: Please provide the main outcome of the present work shortly in this section.

P2, L56-58: There are some other type of waste which can be problematic to the environment. Please take a closer look at the following reference: Ahlatci et al., 2017.

P2, L69-73: Please state clearly the originality and core objectives of this study.

This is a great paper!

As a result, I believe that this paper deserves publication in International Journal of Sustainability through aforementioned minor revisions.

References

Ahlatci, F., Yilmaz, E., Yazici, E.Y., Celep, O., Deveci, H., 2017. Environmental characterization of mineral processing tailings. In: International Symposium on Mining and Environment (ISME), Mugla, Turkey, 27-29 September, pp. 957-967.

Author Response

P1, L25: Please provide the main outcome of the present work shortly in this section.

Suggested changes were introduced to the summary.

P2, L56-58: There are some other type of waste which can be problematic to the environment. Please take a closer look at the following reference: Ahlatci et al., 2017.

Literature indicated by the reviewer (Ahlatci, F., Yilmaz, E., Yazici, E.Y., Celep, O., Deveci, H., 2017. Environmental characterization of mineral processing tailings. In: International Symposium on Mining and Environment (ISME), Mugla, Turkey, 27-29 September, pp. 957-967) is very interesting and valuable and will certainly be used in following researches and articles. However, this work is focused on problems related to fly ashes and bottom ashes which are generated in the process of thermal degradation of municipal wastes because this is a large environmental and economic problem due to monopolization of the waste management depended on the prices of the plan system. For this reason our work deals only with this type of wastes. However, we are very grateful for indicating possible directions for further research.

 P2, L69-73: Please state clearly the originality and core objectives of this study.

The purpose of the research was more detailed and its eco-friendly and original character was emphasized.

The purpose of the performed research was to design an environment friendly concrete mix and to use ashes as an alternative aggregate in accordance with the idea of the Circular Economy.  Performed tests proved that the high resistance of the concrete makes it possible to use this type of waste for example for protecting waste landfill where it may form a layer that separates the waste from the environment. Furthermore, the suggested solution will contribute to the decrease of greenhouse gases, in particular CO2, as the cement is replaced with waste. Moreover, the generated waste will be used in their place of origin which will eliminate economic and environment costs of international transport.

Reviewer 2 Report

This research investigated the immobilization of hazardous fly ashes by analyzing the leaching of contaminants from monolithic and crushed concretes. Overall, the approach of the study in the manuscript is good and could be useful in the public domain, but the manuscript needs considerable revision to reach the public domain. The manuscript also needs language editing, since it is difficult to understand what the authors are trying to convey in places. Authors are suggested to address following comments in order to make the manuscript suitable for publication.

# Abstract:

Line no 16-17, Please rephrase the sentence suitably.

*The abstract should be rewritten by detailing the aim and concept of the manuscript. The abstract should state briefly the purpose of the research, the principal results and major conclusions.

*The abstract always ends outlining the benefits of the study findings and recommendations as a way forward. The manuscript is missing such 1-2 lines in the abstract.

# Introduction:

*Line no 48-50, References missing, Cite references properly.

* Introduction is very general and need to be elaborative to explore the actual philosophy to design the experiment. The introduction is insufficient to provide the state of the art in the topic. Hypothesis should be given. How this work is different from the available data?

The originality and novelty of the paper need to be further clarified. What progress against the most recent state-of-the-art similar studies was made in this study?

*The introduction of the paper must be extended and reformulated in order to provide a more comprehensive approach.

*The objective of the manuscript was not properly described in the Introduction part.

# Material and methods:

*Line no 106-107,109- Please rephrase the sentences.

# Results and discussion:

*Line no 164- should be Results and discussion.

*The manuscript does not provide interesting and technically sound discussion; it would be better to use more recent references in discussion.

*Under section, discussion, it is recommended to discuss and explain what should be the appropriate policies based on the findings of this study. Also, the results should be further elaborated to show how they could be used for real applications. 

* It is strongly recommended to add a subsection, ‘practical implications of this study,’ outlining the challenges in the current research, future work, and recommendations, before the conclusion

# Conclusion:

Conclusion section should be without any bullets/numbering. 

Authors are suggested to draw major inferences/primary conclusions first quoting the data/results obtained followed by the secondary conclusions/ recommendations reached through the critical analysis/ investigation of the study.

Author Response

# Abstract:

Line no 16-17, Please rephrase the sentence suitably.

*The abstract should be rewritten by detailing the aim and concept of the manuscript. The abstract should state briefly the purpose of the research, the principal results and major conclusions.

*The abstract always ends outlining the benefits of the study findings and recommendations as a way forward. The manuscript is missing such 1-2 lines in the abstract.

The works presents tests of ashes from a Polish combustion plant. Nowadays, the management of ashes poses a big problem related to high concentration of contaminations which constitutes an environmental nuisance (heavy metals, chlorides, sulphates, etc).   The excessive leaching of contaminants disqualifies ashes from the depositing in the landfills for hazardous wastes.

# Introduction:

*Line no 48-50, References missing, Cite references properly.

* Introduction is very general and need to be elaborative to explore the actual philosophy to design the experiment. The introduction is insufficient to provide the state of the art in the topic. Hypothesis should be given. How this work is different from the available data?

The originality and novelty of the paper need to be further clarified. What progress against the most recent state-of-the-art similar studies was made in this study?

*The introduction of the paper must be extended and reformulated in order to provide a more comprehensive approach.

*The objective of the manuscript was not properly described in the Introduction part.

The introduction cites material which bring additional information on the stabilization process and literature sources. The introduction has been extended to cover information on the purpose of the research and its original scope.

# Material and methods:

*Line no 106-107,109- Please rephrase the sentences.

The sentence has been reworded, as follows:

For the research,, fly ashes (Fig.2a) and bottom ashes (Fig.2b) have been used. The wastes are generated in the process of thermal degradation of municipal wastes. In accordance with the binding Waste Catalogue [11] the tested ashes (fly and bottom) are classified under code 19 01 07* - Solid wastes from gas treatment. Tested ashes due to the high content of substances harmful to the environment are classified as hazardous waste.

# Results and discussion:

*Line no 164- should be Results and discussion.

*The manuscript does not provide interesting and technically sound discussion; it would be better to use more recent references in discussion.

*Under section, discussion, it is recommended to discuss and explain what should be the appropriate policies based on the findings of this study. Also, the results should be further elaborated to show how they could be used for real applications.

* It is strongly recommended to add a subsection, ‘practical implications of this study,’ outlining the challenges in the current research, future work, and recommendations, before the conclusion

The name of the point has been changed in accordance with the reviewer’s suggestion.

The results section contains additional original information on the mechanical properties of the designed concrete matrixes and shows a list of  applicable results unpublished so far by the authors of the article.

# Conclusion:

Conclusion section should be without any bullets/numbering.

Authors are suggested to draw major inferences/primary conclusions first quoting the data/results obtained followed by the secondary conclusions/ recommendations reached through the critical analysis/ investigation of the study.

Obtained results confirm that tested ashes, which come from the thermal degradation of municipal wastes may be neutralized successfully in concrete matrix. The received test results show that the designed concrete matrix makes it possible to decrease leaching of chlorides and sulphates to the permissible values required by law. The obtained results give basis for the use of the designed concrete matrix with the admixture of ashes for the construction of roads and other construction works on the landfills. Particular attention shall be given to the need of individual preparation of the concrete mix composition depending on the characteristics of the waste in order to achieve the most environment friendly matrix.

Round 2

Reviewer 2 Report

The changes should be marked in the revised manuscript. It is difficult to review the revised manuscript with out showing changes in color. Authors should also explain how and where each point of the reviewers' comments has been incorporated in the revised manuscript. The conclusion part is still with bullets. 

Author Response

The authors introduced changes suggested by the Reviewer. The authors thank the Reviewer for a constructive formulation that has improved the quality of the article.

The authors submitted the article for language correction to a translator specializing in the subject. The article has gone through the language check.

The authors have introduced important changes to the article’s summary (L14-32). The introduction contains information on the realization of the immobilization process. They identified groups of wastes which may be efficiently neutralized by cementation. (L67-95). The authors introduced changes to the purpose of the article specifying its basic assumptions and expected results. (L98-109).  It needs to be added that the authors reviewed carefully literature and found out that there are no identical or similar researchers concerning management of ashes that come from the thermal processing of municipal waste.  (L415-416, L423-438) The subject of new and requires further researches considering in particular possible ways of use of the generated product in the construction industry in accordance with the assumptions of the Circular Economy.

The authors introduced the required changes to the description of the tested material. (L202-208). The authors introduced to the article their own unpublished researches (L327-350). They also added procedures which were the basis for their research. (L202-208). The presented original research shows possibilities to use ashes as a component of cement mortars and concretes. According to the authors, the obtained test results are promising and form a good starting point for further research. Additional argument that supports the need to search for new solution is the response from the industry. At the moment, the authors are cooperating with two large municipal waste thermal degradation plants.

The authors of the article introduced changes to the summary, in accordance to the suggestion of the reviewer. (L359-382).

All the changes were marked in orange colour in the attach manuscript.

The authors are grateful to the reviewer for the constructive remarks which help to prepare a good article and which serve further scientific research.

Round 3

Reviewer 2 Report

The authors have addressed all the comments with full justification. Hence, the paper may be accepted in its current form